# Improving Visual Comfort during Computer Gaming with Preservative-Free Hyaluronic Acid Artificial Tears Added to Ergophthalmological Measures

Fernando Trancoso Vaz [1], Ester Fernández-López [2], María José Roig-Revert [2], Alicia Martín [2] and Cristina Peris-Martínez [2,3,*]

[1] Department of Ophthalmology, Hospital Professor Doutor Fernando Fonseca, 2720-276 Amadora, Portugal
[2] Cornea and Anterior Segment Diseases Unit, FISABIO Medical Ophthalmology, 46015 Valencia, Spain
[3] Department of Surgery, University of Valencia, 46010 Valencia, Spain
* Correspondence: peris_crimar@gva.es

**Abstract:** Digital asthenopia (DA) or Computer Vision Syndrome can occur after prolonged use of digital devices and is usually managed with ergophthalmological measures and the use of artificial tears. This prospective, controlled study evaluated the use of hyaluronic acid artificial tears on the signs and symptoms of DA in participants of a videogame convention. Subjects ($n = 56$) were randomized into a control group (CG, $n = 26$), which followed ergophthalmological measures, and a study group (SG, $n = 30$), which followed ergophthalmological measures and instilled 1 drop of artificial tears with hyaluronic acid 0.15% four times a day. Subjects were evaluated before and after playing for three consecutive days for eye dryness (SPEED questionnaire), conjunctival hyperemia, corneal fluorescein staining, conjunctival lissamine green staining, tear breakup time, Schirmer I test, near convergence and accommodation, and using questionnaires for DA symptoms. After 3 days of intense videogaming, the SPEED score of CG increased significantly ($p = 0.0320$), while for the SG it was unchanged. Similarly, the CG presented significant increases in ocular fatigue ($p = 0.0173$) and dryness ($p = 0.0463$), while these parameters decreased significantly in the SG ($p = 0.0149$ and $p = 0.00427$, respectively). This study confirms the protective effect of hyaluronic acid artificial tears against DA symptoms associated with prolonged visual display terminal use.

**Keywords:** artificial tears; digital asthenopia; computer vision syndrome; dry eye; ergophthalmological measures; eye fatigue; digital eye strain

## 1. Introduction

Digital asthenopia (DA) or Computer Vision Syndrome has been defined as ocular problems related to prolonged use of digital devices such as computers, tablets, e-books and mobile phones [1–9]. DA is a transient and nonspecific disorder that includes ocular, visual, and musculoskeletal (neck and shoulder pain) symptoms [2,10]. There is a significant correlation between the increase in hours of digital screens' use and the onset of symptoms such as burning sensation, blurred vision, and dry eye [10–12]. The symptoms of DA may affect more than half of the people who spend more than 3 or 4 h a day in front of a digital screen and may have a significant economic impact [13,14]. In Spain, the VII National Survey of Working Conditions conducted in 2011 indicated that 77.6% of medical visits in occupational health are attributed to visual complaints related to the use of digital screens [15]. A survey found a high prevalence of DA among university students in Spain [16]. Likewise, a recent report showed that 65% of American adults experience some sort of digital eye strain after prolonged use of electronic devices [12]. A Portuguese ergophthalmology survey and numerous other studies have shown a relationship between prolonged use of digital screens and the appearance of subjective symptomatology and ocular surface changes, suggesting an improvement with the use of ergophthalmological

measures and the use of tears [17–20]. The COVID-19 pandemic has further exacerbated the problem of excessive screen time, as work-related and social meetings and classrooms shifted online [21–23].

There is a relationship between the use of computers and digital devices and dry eye symptoms [10,24]. The overall prevalence of dry eye in computer users was estimated at 49.5% (range 9.5–87.5%) in a recent meta-analysis [25]. A study found that the daily use of >4 h of computer screens was associated with an increased risk of dry eye disease (odds ratio = 1.68; 95% confidence interval = 1.40–2.02) [26]. Computer use affects blink pattern function, with blink frequency decreased by up to about 50% and a higher prevalence of incomplete blinks [12,27]. A reduced blinking frequency is related to higher tear evaporation, and incomplete blinking disrupts the normal secretion of tear components such as the lipid layer and thus contributes to the appearance of symptomatic dry eye, resulting in symptoms such as redness, burning, stinging, or blurred vision [28]. In addition, there are other complaints in DA related to visual strain secondary to near vision effort (accommodation effort and related complaints) [20]. DA is more pronounced in activities such as computer gaming, in which the intensity of the attention required and difficulty of the task further decrease the frequency of the blink rate and can worsen dry eye symptoms [28,29].

Artificial tears and ergophthalmological measures are commonly recommended, based on current research, to reduce the ocular symptoms of DA [9,12,17]. Hyaluronic acid artificial tears have been shown to be effective in relieving dry eye symptoms and improving tear film stability [30]. However, the effectiveness of artificial tears in reducing DA is still uncertain, and few studies have addressed this issue directly [4,17–19]. Therefore, the evaluation of the short-term therapeutic utility of artificial tears combined with ergophthalmological measures through a controlled clinical study in a group of subjects at risk for the development of DA is of interest.

The aim of this study was to evaluate the efficacy of ergophthalmological measures combined with preservative-free hyaluronic acid artificial tears on the short-term management of the signs and symptoms of dry eye and DA before and after prolonged digital screen use.

## 2. Materials and Methods

This prospective, controlled, investigator-masked study enrolled adults participating in DreamHack, a videogame convention held in Valencia, Spain, 4–7 July 2019. The study protocol was approved by the Ethics and Clinical Research Committee of the General University Hospital of Valencia and FISABIO Oftalmología Médica (FOM), and all procedures complied with the Ethical International Standards of Good Clinical Practice (CPMP/ICH/135/95) and the Declaration of Helsinki. All participants provided written informed consent.

### 2.1. Subjects

Subjects included were ≥18 years of age who planned to play videogames for ≥6 h/day for at least 3 consecutive days, considered at risk of DA [26]. Subjects were excluded if they had used any ocular medication in the previous 2 weeks and presented active eye, nasolacrimal, or eyelid disease other than dry eye that required, or not, topical eye treatment; history of trauma or eye infection in the previous 3 months; refractive surgery; near visual acuity (VA) of 0.5 or less in at least one of the two eyes; systemic conditions associated with dry eye (e.g., Sjögren syndrome, rheumatoid arthritis); systemic medication capable of inducing dry eye in the last 30 days; and pregnant, lactating, or non-contraceptive-taking women.

### 2.2. Study Design

The study was conducted during 3 days of the 4-day videogame convention. On Day 1 (D1), the investigator evaluated inclusion and exclusion criteria of the participants, obtained

informed consent, and collected demographic data. The included subjects completed a gaming habits questionnaire, the Standard Patient Evaluation of Eye Dryness (SPEED) questionnaire, and a symptoms questionnaire from 0 to 10 to assess ocular fatigue, blurred vision, burning or ocular pain, ocular dryness, and ocular discomfort. All questionnaires were self-administered. Then, they were evaluated by an ophthalmologist for conjunctival hyperemia (McMonnies Chapman Davies, scale 0–5) [31]; corneal fluorescein staining (Oxford scale 0–5); conjunctival lissamine green staining (a lissamine green strip [I-Dew Green, Entod Research Cell UK Ltd., London, UK] was impregnated with 1 drop saline solution for one minute and applied to the lower fornix, and staining evaluated 2 min after installation, measured with the Oxford scale 0–15); tear breakup time (by measuring with a chronometer the time lapsed until tear breakup after instillation of 1 drop of unpreserved liquid fluorescein and asking the subject to blink 3 times and then cease to blink); Schirmer I test (strip was placed in the temporal 1/3 of the lower lid margin without anesthetic instillation, eyes were closed, and the strips left for 5 min measured with a chronometer); and near point of convergence and accommodation measured in centimeters with a RAF ruler. On Day 3 (D3), after 3 days of videogaming, the subjects completed the SPEED questionnaire and the symptoms questionnaire and underwent an ophthalmologic examination (same as in Day 1). On this visit, the investigators were masked to the treatment received by the subjects.

Subjects were randomized into two groups: the control group (CG) and the study group. The CG followed ergophthalmological measures (standard of care) and did not instill any tear substitute, as the use of any ophthalmological solution would have had an impact, however small, in eye hydration. The SG implemented ergophthalmological measures and instilled 1 drop of preservative-free sterile solution with hyaluronic acid 0.15% (Hyabak®, Laboratoires Théa, Cedex, France) in the conjunctival sac of both eyes 4 times a day during the 3 days of the video gaming session. Adherence to the treatment with artificial tears was estimated by weighing the bottles at Day 1 and Day 3 and comparing to the mean weight of a bottle if 24 drops were used (0.76 mg). It has been suggested that personalized rule breaks, adapted to specific habits of computer users, could be the most beneficial approach [32]. Hence, a suitable ergophthalmological measure was recommended for gamers who cannot stop the game to introduce a pause [33]: participants in both groups were advised to avoid near vision for 1 min, 4 times a day, during the 3 days of the study. Periodic reminder messages for treatment administration were sent to the participants in the SG. When introducing a pause in the SG to insert the drop, a pause was also introduced in the CG so that the circumstances were similar, and to control for differences due to the introduction of the tear substitute.

### 2.3. Endpoints and Assessments

The primary endpoint was the change in the SPEED questionnaire score between D1 and D3 (range: 0 to 28) [34], comparing the SG and the CG.

The secondary endpoints of this study included the evaluation and comparison between D1 and D3 of scores of conjunctival hyperemia assessed with the modified version of the McMonnies/Chapman-Davies scale [31]; corneal fluorescein staining (CFS); corneal and conjunctival lissamine green (CLS) staining; tear breakup time (TBUT); Schirmer I test; near visual acuity (NVA); and near point of convergence and accommodation (NPC and NPA, respectively). Likewise, a symptomatic questionnaire using Visual Analog Scales (VAS) was used on D1 and D3. Subjective ocular comfort after instillation of the artificial tears was evaluated for the SG at D3.

### 2.4. Sample Size Calculation

The sample size calculation was based on published data of mean SPEED questionnaire scores of healthy people (5.24 ± 3.04 units) [35] and patients with symptomatic dry eye (9 ± 4 units) [36]. Considering an alpha error of 5% and a beta error < 20%, and defining as a patient with evaporative dry eye one who obtained a SPEED score of at least 9 units, it

was estimated that 25 patients were needed per group (50 in total) to detect statistically significant differences.

*2.5. Statistical Analysis*

Data analyses were performed using the SAS statistical package (Statistical Analysis System, version 9.2, SAS Institute Inc., Cary, NC, USA). A descriptive analysis of all variables collected in the study were performed and stratified per visit. Continuous variables were described in terms of a central tendency (mean, standard deviation [SD], minimum–maximum, confidence interval [CI] 95%), and the categorical or ordinal variables were described by frequencies (*n*) and percentages (%).

For the primary and the secondary endpoints, a comparison of the data and scores obtained at baseline and D3 were performed globally and for each group of treatment, when appropriate. The changes observed in the scores of the different scales and tests among both treatment groups were also compared using T Student tests for paired data (intra-group) and for independent data (inter-group). For ordinal and other variable types, the Wilcoxon signed-rank test for intra-group comparisons and the Mann–Whitney U-test for inter-group comparisons were used. Categorical variables were compared using the Fisher exact test or the Chi Square test. The worst eye from each participant was chosen as the study eye and defined as the eye with higher hyperemia; if both equal, the eye with lower TBUT; if both equal, the eye with higher CLS; and if both equal, the right eye.

For all tests in the study, the significance level was established at *p* < 0.05.

All adverse effects reported by participants regarding the use of the artificial tears were registered and presented through descriptive statistics.

## 3. Results

A total of 56 subjects were randomized to the SG (*n* = 30, 53.6%) and the CG (*n* = 26, 46.4%). The mean ± SD age was 25.2 ± 5.8 years (range: 18–45 years), and 92.9% were men (Table 1). Participants reported an average of 25.0 ± 17.7 h/week of gaming, and most (59.2%) indicated taking breaks during their gaming sessions of a median of 10 min every hour, approximately.

Each study participant reported using, on average, two different platforms for gaming per day, the most frequent being the computer (98.0%) and the mobile phone (73.5%), similar in both groups (*p* = 0.489 and *p* = 1.000, respectively).

There was no statistically significant difference in the SPEED score between D1 and D3 for the SG (*p* = 0.4621), but a significant increase in the CG was observed (*p* = 0.0320) (Figure 1A). The difference in scores between D1 and D3 in both groups was statistically significant (−0.33 vs. 1.54, respectively, *p* = 0.0221).

**Table 1.** Demographic and clinical characteristics and self-reported gaming habits of the participants.

| Variable | SG (*n* = 30) | CG (*n* = 26) | All (*n* = 56) | *p* Value |
|---|---|---|---|---|
| Gender, *n* (%) | | | | |
| Male | 28 (93.3) | 24 (92.3) | 52 (92.9) | 1.0000 [a] |
| Female | 2 (6.7) | 2 (7.7) | 4 (7.1) | |
| Age (years), mean ± SD | 25.3 ± 5.7 | 25.0 ± 6.1 | 25.2 ± 5.8 | 0.8523 [b] |
| Usage of visual aids, *n* (%) | | | | |
| Eyeglasses | 10 (40.0) | 9 (37.5) | 19 (38.8) | 1.0000 [a] |
| Contact lenses | 5 (20.0) | 2 (9.1) | 7 (14.9) | 0.4227 [a] |
| None | 20 (66.7) | 17 (65.4) | 37 (66.1) | 1.0000 [a] |
| Gaming habits | | | | |
| Hours playing/week, mean ± SD | 24.6 ± 14.7 | 25.5 ± 20.6 | 25.0 ± 17.7 | 0.8482 [b] |
| Minutes between breaks, mean ± SD | 87.8 ± 58.1 | 77.1 ± 45.3 | 82.7 ± 51.7 | 0.5883 [b] |
| Break duration (minutes), mean ± SD | 19.1 ± 28.5 | 13.1 ± 13.6 | 16.2 ± 22.4 | 0.4761 [b] |
| "No breaks" players, *n* (%) | 10 (40.0) | 10 (41.7) | 20 (40.8) | 1.0000 [a] |

**Table 1.** *Cont.*

| Variable | SG (*n* = 30) | CG (*n* = 26) | All (*n* = 56) | *p* Value |
|---|---|---|---|---|
| Game Devices, *n* (%) | | | | |
| Computer | 25 (100.0) | 23 (95.8) | 48 (98.0) | 0.4898 [a] |
| Mobile phone | 18 (72.0) | 18 (75.0) | 36 (73.5) | 1.0000 [a] |
| Video console | 8 (32.0) | 10 (41.7) | 18 (36.7) | 0.5607 [a] |
| Portable console | 3 (12.0) | 6 (25.0) | 9 (18.4) | 0.2890 [a] |
| Portable device | 3 (12.0) | 3 (12.5) | 6 (12.2) | 1.0000 [a] |
| Tablet | 2 (8.0) | 3 (8.0) | 5 (10.2) | 0.6671 [a] |
| Arcade machine | 0 (0.0) | 1 (4.2) | 1 (2.0) | 0.4898 [a] |
| Game devices used/day, *n* (%) | | | | |
| 1 device | 4 (16.0) | 2 (10.5) | 6 (13.6) | – |
| 2 devices | 14 (56.0) | 9 (47.4) | 23 (52.3) | |
| 3 devices | 7 (28.0) | 6 (31.6) | 13 (29.6) | |
| 4 devices | 0 (0.0) | 2 (10.5) | 2 (4.6) | |
| Mean ± SD | 2.1 ± 0.7 | 2.4 ± 0.8 | 2.3 ± 0.8 | 0.2522 [c] |

[a] Fisher test, [b] T-test, [c] Mann–Whitney U-test. CG, control group; SD, standard deviation; SG, study group.

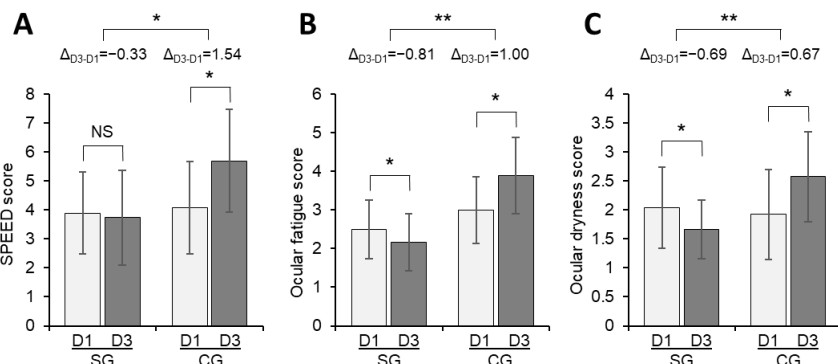

**Figure 1.** Change in the SPEED score (**A**), ocular fatigue score (**B**), and ocular dryness score (**C**) from D1 to D3 for the SG and CG. The bars represent means, and the error bars the 95% confidence interval. Statistical significance: *, $p < 0.05$; **, $p < 0.001$. CG, control group; D1, Day 1; D3, Day 3; ΔD1 −D3, difference between Day 1 and Day 3; NS, not significant; SG, study group.

Concerning the subjective assessment of symptoms by visual analog scales (VAS), the SG showed significant decreases in ocular fatigue ($p = 0.0149$) and ocular dryness ($p = 0.00427$), whereas the CG presented significant increments in both parameters ($p = 0.0173$ and $p = 0.0463$, respectively) (Figure 1B,C).

Neither of the two groups presented significant changes between D1 and D3 in blurred vision score, eye discomfort, or burning and ocular pain (Figure 2). Finally, the subjective questionnaire also reported that 83.3% of the participants in the SG reported their eyes feeling better or much better after instillation of the artificial tears.

No differences were observed in the clinical parameters assessed in the ophthalmic evaluation in D1 and D3 for the SG or the CG (Table 2).

Regarding adherence to the treatment with artificial tears, 77% of the participants extracted from the bottle at least 50% of the drops required, and 53% of participants at least 80% of the drops.

Only two adverse events (AEs) were reported but were not linked to the study treatment according to the researcher: one patient noted a bright yellow ocular secretion after fluorescein instillation, and another experienced faintness during the Schirmer exam but quickly recovered.

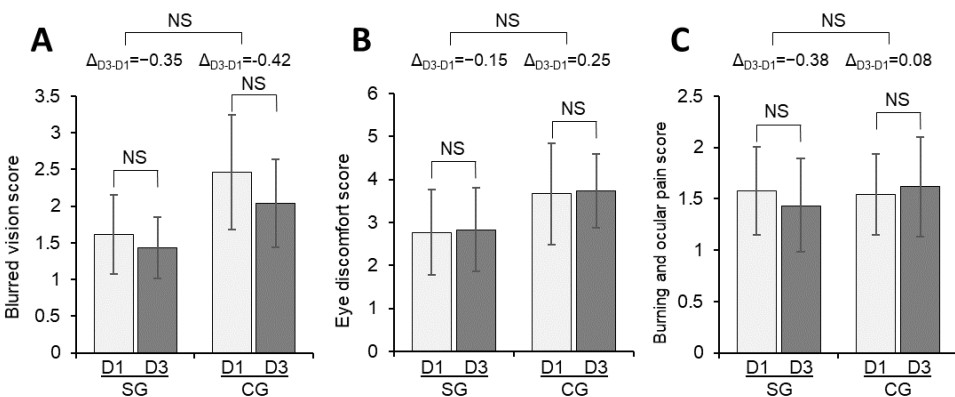

**Figure 2.** Change in blurred vision score (**A**), eye discomfort score (**B**), and burning and ocular pain score (**C**) from D1 to D3 for the SG and CG. The bars represent means, and the error bars the 95% confidence interval. CG, control group; D1, Day 1; D3, Day 3; ΔD1–D3, difference between Day 1 and Day 3; NS, not significant; SG, study group.

**Table 2.** Ophthalmic evaluation of eye signs of digital asthenopia. For the SG, N = 60 eyes; for the CG, N = 52 eyes.

| Variable | Day 1 | Day 3 | $p$ Value [1] |
|---|---|---|---|
| Near visual acuity | | | |
| SG | 1.2 ± 0.3 | 1.3 ± 0.3 | 0.2168 |
| CG | 1.2 ± 0.3 | 1.2 ± 0.3 | 0.7492 |
| Near point of accommodation | | | |
| SG | 12.9 ± 4.2 | 12.4 ± 3.2 | 0.2153 |
| CG | 11.7 ± 4.2 | 12.0 ± 4.1 | 0.2349 |
| Near point of convergence | | | |
| SG | 8.4 ± 2.9 | 7.8 ± 3.3 | 0.3939 |
| CG | 9.0 ± 5.2 | 8.2 ± 5.2 | 0.3340 |
| Conjunctival hyperemia | | | |
| SG | 0.3 ± 0.5 | 0.3 ± 0.6 | 0.7136 |
| CG | 0.1 ± 0.3 | 0.4 ± 0.7 | 0.0828 |
| TBUT (seg) | | | |
| SG | 6.7 ± 3.7 | 5.6 ± 3.1 | 0.0759 |
| CG | 7.0 ± 3.2 | 6.7 ± 3.7 | 0.1047 |
| Corneal fluorescein staining | | | |
| SG | 0.4 ± 0.6 | 0.4 ± 0.7 | 0.8012 |
| CG | 0.2 ± 0.4 | 0.3 ± 0.5 | 0.2649 |
| Corneal and conjunctival lissamine green staining | | | |
| SG | 0.3 ± 0.6 | 0.2 ± 0.4 | 0.3746 |
| CG | 0.2 ± 0.4 | 0.2 ± 0.4 | 0.6636 |
| Schirmer's Test | | | |
| SG | 20.8 ± 10.3 | 21.9 ± 11.5 | 0.5263 |
| CG | 18.7 ± 10.4 | 20.2 ± 12.5 | 0.4464 |

[1] Paired T-Test. All values are given as mean ± SD. CG, control group; SG, study group; TBUT, tear breakup time.

## 4. Discussion

This study evaluated the use of ergophthalmological measures (standard of care) with the use or not of preservative-free hyaluronic acid artificial tears in DA complaints. The results provide evidence of the beneficial effect of artificial tears for the management of ocular dryness symptoms in healthy young video gamers, in combination with regular breaks (ergophthalmological measures). A significant increase in ocular symptoms of dry eye evaluated with the SPEED questionnaire was found in the CG, whilst the use of artificial tears in the SG avoided this effect. Mean symptom scores related to the secondary assessment using VAS also showed a positive effect of artificial tears on ocular fatigue and eye dryness. The lack of adverse events related to the study treatment confirmed the

safety and good tolerability of the hyaluronic acid artificial tears studied. These results observed in the CG were consistent with other studies, which showed that symptoms such as burning sensation, blurred vision, vision discomfort, and dry eye are common among video intensive users of digital displays [10–12,29,37]. In this study, changes in the signs of dry eye were observed, although not statistically significant, probably due to the short follow-up period of only three days. The artificial tears used in the study were preservative-free to avoid the risk of adverse reactions to preservatives on the ocular surface (e.g., keratopathy caused by the excipient benzalkonium chloride) [38]. If adverse events caused by preservatives persist, they may have a negative impact on adherence [39].

Artificial tears have been traditionally used for the treatment of dry eye disease to improve symptoms [9,12,17]. Hyaluronic acid, included in the formulation of the artificial tears used in this study, occurs naturally in the human body. It increases viscosity of the tear film, improves retention time, optimizes ocular surface hydration and lubrication, and may be useful to treat conditions where epithelial recovery is necessary [40]. Another study evaluated the effect of recommending hyaluronic artificial tears, environmental measures, and introducing pauses during computer activity [20]. After 1 month, a decrease in eye fatigue and improvements in parameters such as the Schirmer test, lacrimal film, keratitis, conjunctival lesions, and near convergence point was found. Consistent with other studies, the major visual symptom reported by participants at the beginning of the study was ocular fatigue [29]. A reduction in ocular fatigue was observed after the use of artificial tears during intensive use of the visual display terminal in gaming sessions. This significant improvement in digital asthenopia symptoms (ocular fatigue, eye dryness, or eye discomfort) is important given the impact of dry eyes on quality of life [41]. Although symptoms are usually temporary, the effect on eye fatigue should be especially emphasized, as it could be a potential factor for decreased productivity associated with prolonged use of digital screens [29,42].

Dry eye disease is defined as is a multifactorial disease of the ocular surface characterized by a loss of homeostasis of the tear film and accompanied by ocular symptoms, in which tear film instability and hyperosmolarity, ocular surface inflammation and damage, and neurosensory abnormalities play etiological roles [43]. Short breakup time dry eye (SBUDE) is a symptomatic subtype of dry eye disease, with a fluorescein breakup time of $\leq 5$ s, occurring in the presence of a normal tear secretion and clearance and normal meibomian gland function and unassociated with epithelial damage [44,45]. Most of the patients in our study had a short TBUT and a normal Schirmer, so it could be assumed that when symptomatic, their symptoms were due to an evaporative dry eye of the SBUDE type.

The occurrence of symptomatic dry eye in users of digital screens has been related to a reduction of blink rate frequency, incomplete blinking, and also to a greater tear evaporation with stare. Tasks that are difficult and demand high attention induce the user to stare at the screen for long periods of time, leading to tear film instability [10,27,28]. In addition, an increased effort in near vision is associated with an increased effort in accommodation/convergence. When this effort is pronounced and/or maintained (>2 h/day), failure of adaptation mechanisms can occur, such as exhaustion of the ocular muscles (intrinsic and extrinsic) and subsequent visual fatigue (asthenopia) [20]. The population included in our study was exposed to intense and prolonged use of digital screens, with tasks that require high concentration and difficulty.

Although it is commonly accepted that artificial tears should be instilled regularly throughout the day for patients with dry eye or at risk of developing dry eye to prevent symptoms, an approach has been proposed with a dynamic therapeutic strategy according to clinical symptoms [40]. Changes in the environment can contribute to the development or worsening of dry eye symptoms. Patients should monitor their condition and modify the frequency and/or type of eye drops according to a change in symptoms. This approach would be the most appropriate for the healthy population that uses digital devices, as the daily duration of digital screen use and the risk of developing dry eye symptoms can be highly variable [40].

Some limitations of this study should be considered. This was a short-term assessment of the changes in a context of heavy visual display terminal use, as opposed to long-term exposure of office workers who use a computer every day for long hours, although probably with less attention to the screen. After only 3 days of the intervention, only small changes could be expected, even if the exposure to digital screens was intensive. However, this fact makes the small significant changes observed even more relevant. Additionally, in this study, environmental factors such as screen light intensity, ventilation, smoking, and alcohol intake were not controlled. However, as the development of the study and the gaming activity were carried out in parallel in the same room of the convention center, it can be assumed that the conditions were identical for both groups and that it provides real world evidence on what to expect from gamers. The CG used only ergophthalmological measures, as any ophthalmic solution could have had an undetermined effect in eye hydration that would have made the interpretation of the results more complicated. In addition, adherence to the recommendation of taking breaks could have been lower than adherence to artificial tear instillation. Another limitation was that the period that elapsed between the gaming sessions and the ophthalmological evaluations was not controlled. Parameters such as the TBUT can quickly reverse to normal ranges if blinking rates are reestablished after gaming, as shown previously [28]. In this regard, it should be noted that the average TBUT values observed were below normal (>10 s is considered normal), which is unexpected in the healthy young population included in the study. Finally, it should be noted that measuring hyperemia with the McMonnies scale is a limitation of the study because it is a subjective measure.

## 5. Conclusions

The findings herein suggest that preservative free hyaluronic acid artificial tears, and concomitant use of ergophthalmological measures such as regular breaks, are well tolerated and could have a protective effect against dry eye symptoms associated with prolonged and/or intensive visual display terminal use.

**Author Contributions:** Conceptualization, F.T.V., E.F.-L. and C.P.-M.; Data curation, F.T.V., E.F.-L., M.J.R.-R. and C.P.-M.; Investigation, F.T.V., E.F.-L., M.J.R.-R and A.M.; Methodology, F.T.V., E.F.-L. and C.P.-M.; Validation, F.T.V., E.F.-L. and C.P.-M.; Writing—original draft, F.T.V., E.F.-L. and C.P.-M. All authors have read and agreed to the published version of the manuscript.

**Funding:** This study was funded by Laboratoires Théa SA.

**Institutional Review Board Statement:** The study was conducted in accordance with the Declaration of Helsinki and approved by the Institutional Review Board of the General University Hospital of Valencia and FISABIO Oftalmología Médica (FOM).

**Informed Consent Statement:** Informed consent was obtained from all subjects involved in the study.

**Data Availability Statement:** The datasets generated during and/or analyzed during the current study are available from the corresponding author on reasonable request.

**Acknowledgments:** The authors thank Francisco López de Saro (Trialance SCCL) for medical writing support, funded by Laboratoires Théa SA.

**Conflicts of Interest:** The authors declare no conflict of interest with respect to the research, authorship, and/or publication of this article.

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
