# Peer review of "Improving Visual Comfort during Computer Gaming with Preservative-Free Hyaluronic Acid Artificial Tears Added to Ergophthalmological Measures"

_2411-5150, 2005_

Round 1
Reviewer 1 Report
This study evaluated the use of ergophthalmological measures (standard of care) with the use or not of preservative-free hyaluronic acid artificial tears, in DA complaints. The results provide evidence of the beneficial effect of artificial tears for the management of ocular dryness symptoms in healthy young video gamers, in combination with regular breaks (ergophthalmological measures). A significant increase in ocular symptoms of dry eye evaluated with the SPEED questionnaire was found in the CG, whilst the use of artificial tears in the SG avoided this effect.
I have only minor comments: please find other publications showing visual changes in heavy computer gamers and add to introduction, disussion and referenes sections.
Author Response
Reviewer #1
This study evaluated the use of ergophthalmological measures (standard of care) with the use or not of preservative-free hyaluronic acid artificial tears, in DA complaints. The results provide evidence of the beneficial effect of artificial tears for the management of ocular dryness symptoms in healthy young video gamers, in combination with regular breaks (ergophthalmological measures). A significant increase in ocular symptoms of dry eye evaluated with the SPEED questionnaire was found in the CG, whilst the use of artificial tears in the SG avoided this effect.
I have only minor comments: please find other publications showing visual changes in heavy computer gamers and add to introduction, disussion and referenes sections.
RESPONSE: Thank you for your suggestion. We have now added new references in the Introduction and Discussion (references 1-20), related to prior research on the topic of digital asthenopia. These references cover studies from 2005 to 2021.

Reviewer 2 Report
This is an interesting study, but lacks methodological detail and the limitations are not fully discussed.
Digital eye strain would be an important Keyword
Ln46 The VIII National Survey of Working Conditions is over a decade old – can a more modern reference be used? Ergophthalmological doesn’t seem to be a well-established term so needs careful definition. In this study it seems to be just recommending a pause. The 2019 study is not “ the first study of digital asthenopia that includes both a subjective (questionnaire) as well as an objective assessment” as stated e.g. Chatterjee P et al., (2005) Comparative randomised active drug controlled clinical trial of a herbal eye drop in computer vision syndrome. Journal of the Indian Medical Association 2005;103: 397-8 amongst many others). This study [12] also had a range of possible interventions explained to the treatment group and compliance was not assessed.
Ln69-70 ergophthalmological measures are not standard, but are perhaps “recommended based on current research”
Ln72 references to date of artificial tear treatment studies in digital environments need to be cited and critiqued.
Ln75 there are no inclusion criteria stated so how are the study group “at risk for the development of DA”?
Ln82 “blinding” is not appropriate terminology in ophthalmology. Please use “masking” and state who was masked – investigator or participants.
Ln105-6 identify how each of these tests were performed, such as were the questionnaires self-administered? Which lissamine green was used and were the TFOS DEWS II recommendations followed etc – Ln130-137 should be relocated here and expanded upon
Ln119 what evidence was the 1 minute 4x a day based on?
Ln119-123 How was compliance measured and more details of the form and frequency of the reminders needs to be articulated.
Ln129 The McMonnies/Chapman-Davies scale has been surpassed by better validated hyperaemia scales. Why was this used and it should be stated as a limitation.
Ln135 why were the VAS scales needed as well as the SPEED?
Ln152-164 it is unclear which statistical tests were used for each of the parameters. As the two time points are with the same person, a repeated measures ANOVA with time as the within subject assessment and group as a between subject variable would be more appropriate for continuous data.
Table 1 legend should note the measures such as gaming habits are self-reported
What are “other” vision rated pathologies. Refractive error is generally not considered a pathology. The current analysis doesn’t account for the degree of refractive error and if self-reported has limited credibility.
Ln202 the adverse events should be described.
In 228 Does “these results” relate to the control group?
Ln231 insert a comma after “In this study . . .”
Ln245 In study [12] an artificial tear was one of many methods to alleviate DA suggested to patients and compliance was not checked, so this sentance is overstated
Ln254-5 This is (half of) the old (2007) definition of dry eye. Please use the 2017 TFOS DEWS II definition
Ln274 it is not clear whether the proposal is coming from the researchers or others (who should be cited)
Ln281 There is a strong risk of bias that providing artificial tears enhanced compliance over fairly basic ‘take breaks’ advice. This needs to be clearly articulated and the study conclusions / abstract adjusted accordingly
Author Response
- This is an interesting study, but lacks methodological detail and the limitations are not fully discussed. Digital eye strain would be an important Keyword
RESPONSE: Thank you for the suggestion. We have added “Digital eye strain” in the keyword list.
- Ln46 The VIII National Survey of Working Conditions is over a decade old – can a more modern reference be used? Ergophthalmological doesn’t seem to be a well-established term so needs careful definition. In this study it seems to be just recommending a pause. The 2019 study is not “ the first study of digital asthenopia that includes both a subjective (questionnaire) as well as an objective assessment” as stated e.g. Chatterjee P et al., (2005) Comparative randomised active drug controlled clinical trial of a herbal eye drop in computer vision syndrome. Journal of the Indian Medical Association 2005;103: 397-8 amongst many others). This study [12] also had a range of possible interventions explained to the treatment group and compliance was not assessed.
RESPONSE: You are correct. We have reworded this paragraph to include other studies on the topic in addition to the Portuguese study. We have added the Chatterjee study of 2005 and Guillon 2004.
- Ln69-70 ergophthalmological measures are not standard, but are perhaps “recommended based on current research”
RESPONSE: We have reworded as suggested to “recommended based on current research”.
- Ln72 references to date of artificial tear treatment studies in digital environments need to be cited and critiqued.
RESPONSE: We have rewritten the sentence and added references. However, due to space limitations, we cannot review all the studies on the topic here. We reference a recent review on the topic (Mehra & Galor 2020).
- Ln75 there are no inclusion criteria stated so how are the study group “at risk for the development of DA”?
RESPONSE: A previous study had shown that people exposed to >4 hours of computer screens per day was associated had an increased risk of dry eye disease (odds ratio=1.68; 95% confidence interval=1.40-2.02) (Reference 26). In our study, the group included in the study had to play videogames for ≥6 hours/day for at least 3 consecutive days. Therefore, this group could be considered at risk of DA. Inclusion and exclusion criteria can be found in lines 94–102.
- Ln82 “blinding” is not appropriate terminology in ophthalmology. Please use “masking” and state who was masked – investigator or participants.
RESPONSE: Thank you, we made the correction to “investigator-masked”. In the second visit of the study, after 3 days of videogame playing, the investigators were masked to the treatment received by the subjects.
- Ln105-6 identify how each of these tests were performed, such as were the questionnaires self-administered? Which lissamine green was used and were the TFOS DEWS II recommendations followed etc – Ln130-137 should be relocated here and expanded upon
RESPONSE: Yes, the questionnaires were self-administered. Details have been added to the methods used in the ophthalmological examination, lines 107–122, as follows:
Then they were evaluated by an ophthalmologist for conjunctival hyperemia (McMon-nies Chapman Davies, scale 0-5); corneal fluorescein staining (Oxford scale 0-5); con-junctival lissamine green staining (lissamine green strip was impregnated with 1 drop saline solution and applied to the lower fornix and staining evaluated 2 minutes after installation, measured with the Oxford scale 0-15); tear breakup time by measuring with a chronometer the time lapsed until tear breakup after instillation of 1 drop of un-preserved liquid fluorescein and asking the subject to blink 3 times and then cease to blink; Schirmer I test (strip was placed in the temporal 1/3 of the lower lid margin without anesthetic instillation, eyes were closed and the strips left for 5 minutes measured with a chronometer); and near point of convergence and accommodation measured in centimeters with a RAF ruler.
- Ln119 what evidence was the 1 minute 4x a day based on?
RESPONSE: There is no prior evidence. As the intervention group would have to make a pause to instill the artificial tears 4 times a day, the control group was asked to make a pause that would be equivalent.
- Ln119-123 How was compliance measured and more details of the form and frequency of the reminders needs to be articulated.
RESPONSE: All participants had a reminder card in front of their screen, mobile messages were sent to the participants reminding to instill the drops and take a pause. Tear drop compliance was measured weighing the bottles before and after and 100% compliance assumed if bottle weight decreased by 0.76 mg which is the mean weight the bottles lose after instilling 24 drops. We have now added a sentence in Methods as follows (lines 129–132): “Adherence to the treatment with artificial tears was estimated by weighing the bottles at Day 1 and Day 3 and comparing to the mean weight of a bottle if 24 drops were used (0.76 mg).” We have also added the following assessment in Results (lines 213–216): “Regarding adherence to the treatment with artificial tears, we found that 77% of the participants extracted from the bottle at least 50% of the drops required, and 53% of participants at least 80% of the drops.”
- Ln129 The McMonnies/Chapman-Davies scale has been surpassed by better validated hyperaemia scales. Why was this used and it should be stated as a limitation.
RESPONSE: The McMonnies scale was used because it is a scale used routinely in clinical studies in ophthalmology and has been validated. We have added in the Discussion that measuring hyperemia with the McMonnies scale was a limitation of the study because it is a subjective measure.
- Ln135 why were the VAS scales needed as well as the SPEED?
RESPONSE: Our goal was to discriminate more precisely what symptoms were modified and to what degree. As there were only 3 days of gaming, we expected small changes that could go undetected with a Likert scale. Also, the SPEED score mixes together sets of symptoms: dryness+foreign body sensation; pain+irritation; burning+tearing, then ocular fatigue. To discriminate what symptoms gamers manifest we included the VAS scale.
- Ln152-164 it is unclear which statistical tests were used for each of the parameters. As the two time points are with the same person, a repeated measures ANOVA with time as the within subject assessment and group as a between subject variable would be more appropriate for continuous data.
RESPONSE: Although it is true that the model you propose would be the most suitable for carrying out all the analyzes together, the analysis carried out is totally equivalent. For continuous variables, firstly, the effect within each treatment has been evaluated, using a T-Test for paired data, and to evaluate the effect between treatments, a T-Test of the difference between the previous evaluation has been carried out and the subsequent assessment comparing this difference between groups. The final result of the methodology you suggest, and the one we used, is similar.
- Table 1 legend should note the measures such as gaming habits are self-reported. What are “other” vision rated pathologies. Refractive error is generally not considered a pathology. The current analysis doesn’t account for the degree of refractive error and if self-reported has limited credibility.
RESPONSE: Thank you for the suggestion. We have added “self-reported” in the legend. Also, we deleted the self-reported pathologies, as they were not relevant for the study.
- Ln202 the adverse events should be described.
RESPONSE: We added a description of the two adverse events. One patient noted a bright yellow ocular secretion after fluorescein instillation and another experienced faintness during the Schirmer exam that quickly recovered.
- In 228 Does “these results” relate to the control group?
RESPONSE: Added: “The results observed in the CG”… to clarify.
- Ln231 insert a comma after “In this study . . .”
RESPONSE: Added, thank you.
- Ln245 In study [12] an artificial tear was one of many methods to alleviate DA suggested to patients and compliance was not checked, so this sentence is overstated
RESPONSE: We have rewritten the sentence as follows: “Another study evaluated the effect of recommending hyaluronic artificial tears, environmental measures and introducing pauses during computer activity [20]. After 1 month, a decrease in eye fatigue and improvements in parameters such as the Schirmer test, lacrimal film, keratitis, conjunctival lesions and near convergence point was found.”
- Ln254-5 This is (half of) the old (2007) definition of dry eye. Please use the 2017 TFOS DEWS II definition
RESPONSE: Added full definition in 2017 TFOS DEWS II, and reference (Craig 2017).
- Ln274 it is not clear whether the proposal is coming from the researchers or others (who should be cited)
RESPONSE: It is current reference 35. Cited earlier in the paragraph now.
- Ln281 There is a strong risk of bias that providing artificial tears enhanced compliance over fairly basic ‘take breaks’ advice. This needs to be clearly articulated and the study conclusions / abstract adjusted accordingly
RESPONSE: We have added a sentence reflecting this point in the limitations, as follows: “Also, adherence to the recommendation of taking breaks could have been lower than adherence to artificial tear instillation.” We have also reworded the Conclusions.

Round 2
Reviewer 2 Report
Thank you for addressing my comments. Please avoid using personal pronouns "we found that 77%". It would be good to state how long the saline drop was on the lissamine green strip before application and the brand used (as this affects the appearance)
Author Response
Thank you for suggesting the changes. We added the requested details for the conjunctival lissamine green staining as follows: "a lissamine green strip [I-Dew Green, Entod Research Cell UK Ltd, London, UK] was impregnated with 1 drop saline solution for one minute and applied to the lower fornix, and staining evaluated 2 minutes after ..." We also made the proposed correction in line 208.
Thank you,
The authors